# OpenReview forum: "Jailbreaking Jailbreaks: A Proactive Defense for LLMs"
_ICLR.cc/2026/Conference — ICLR 2026 Conference Withdrawn Submission_

### Official Review · Reviewer_kWYP · 2025-10-27

**Soundness:** 3
**Presentation:** 3
**Contribution:** 3
**Rating:** 4
**Confidence:** 4

**Summary:**

The paper proposes PROACT, a proactive defense that “jailbreaks the jailbreak” by returning spurious, non-harmful responses that look like successful jailbreak outputs to an attacker’s evaluator, thereby misleading and prematurely terminating the adversarial search. PROACT is instantiated as a three-agent pipeline: a User Intent Analyzer routes only malicious queries into the defense, a Proactive Defender generates topic-consistent but benign “encoded” outputs that appear harmful, and a Surrogate Evaluator iteratively critiques and refines these spurious responses until they pass a jailbreak evaluator. Across different target LLMs and attack frameworks, PROACT significantly lowers while preserving instruction-following utility.

**Strengths:**

1. The idea of "Jailbreaking jailbreaks" is very interesting and novel.

2, The paper is well-written and easy to follow

3. The empirical result of the PROACT method seems very good, and it can work with other defense methods together.

**Weaknesses:**

1,
The Achilles heel of the PROACT system might be its User Intent Analyser, as it is basically acting as an LLM judge that detects the harmful content in the user input. One ICML 2025 paper (https://icml.cc/virtual/2025/poster/45356) specifically talks about how the LLM detection can be bypassed with their attack. The paper might need to address the usage of the LLM as a User Intent Analyser further.

2.
Some work, such as the persuasion attack (https://doi.org/10.18653/v1/2024.acl-long.773), also use a LLM judge to score the response's harmfulness and I am curious if that will work against the surrogate evaluator.

3,
Some jailbreak attacks do not involve iterative optimizations. For example, long-context jailbreaks (https://arxiv.org/pdf/2402.16717) or many-shot jailbreaking (https://proceedings.neurips.cc/paper_files/paper/2024/hash/ea456e232efb72d261715e33ce25f208-Abstract-Conference.html). Is PROACT defenseless against these types of attacks?

4.
It seems like there are many back and forth between LLMs within the PROACT system. It raises the concern of increasing latency and resources of computing.

**Questions:**

1, How will the PROACT work against the jailbreak attack that was designed to bypass LLM detection (https://icml.cc/virtual/2025/poster/45356), or does not need iterative optimization (https://proceedings.neurips.cc/paper_files/paper/2024/hash/ea456e232efb72d261715e33ce25f208-Abstract-Conference.html)?


2, What is the extra latency of inferencing with PROACT?

---

> ### Author Response · Authors · 2025-11-20
>
> We sincerely thank the reviewer for their detailed and thoughtful comments. We are glad the reviewer notes the novelty and effectiveness of our approach, along with the quality of our presentation.
>
> ## Q1: Intent Analyser ##
>
> We appreciate the reviewer raising this concern and are happy to clarify. While the User Intent Analyzer can be imperfect (average detection rate 76.1% across Table 1, 100% accuracy under utility test), **ProAct relies far less on the analyzer’s detection accuracy than existing defenses**. Iterative jailbreak attacks begin with naive prompts, and each refusal provides feedback that strengthens the attacker’s optimization loop. **Prior defenses are passive and must therefore detect malicious intent at every step, which becomes increasingly difficult as attacks evolve** (see table below). In contrast, ProAct **needs to intervene only once when attacks are still weak**, generating a spurious response that stops the optimization loop entirely and preventing long iterative attacks.
>
> Table 2 confirms this. Across four benchmarks and two strong attack strategies, ProAct reduces ASR by 33.25% beyond input filters and 12.38% beyond output filters because it requires only one successful detection. Against DAGR, which explicitly obfuscates intent, ProAct still provides up to 23% additional reduction. Regarding the referenced ICML work, the Emoji Attack manipulates the base model’s harmful outputs before they are sent to a post-screening filter, but **ProAct stops adversarial prompts before any model output is generated**, effectively countering this approach.
>
> ProAct is also **fully compatible with existing defense pipelines**, and the User Intent Analyzer **can consist of signals from multiple other defense mechanisms**. Layering ProAct with AutoDefense in this manner reduces X-Teaming ASR to 0%, and combining it with other baselines yields up to an additional 70% reduction. Imperfections in any single analyzer are mitigated, since a **single positive detection from any component** triggers ProAct and prevents the jailbreak. This makes ProAct practical to deploy as a proactive layer in real-world LLM safety systems.
> We will include this discussion in the final version of our paper.
> |Iteration|Percentage|
> |:--|:--|
> |step 1|12%|
> |step 2|26%|
> |step 3|36%|
> |step 4|80%|
> |step 5|100%|
>
> Table: Cumulative Percentage of Successful PAIR Attacks (out of 100 random samples) against an Input Filter at each Optimization Step
>
> ## Q2: ProAct against Persuasion Attack’s Judge ##
>
> We thank the reviewer for this clarification. We sampled 100 spurious ProAct responses against PAIR on AIR-Bench, pairing them with malicious goals for GPT-Judge evaluation (with GPT-4.1-mini and GPT-5).
> The judge outputs a discrete score from 1 (non-compliance) to 5 (full, detailed compliance). ProAct consistently achieved high scores under both judge configurations. This demonstrates that **our spurious responses remain highly effective when assessed by this advanced and carefully crafted judging framework**.
> |model| score|
> |---|---|
> |GPT-4.1-mini|4.8|
> |GPT-5|4.52|
>
> ## Q3: Long-context Jailbreak and Many Shot Jailbreak ##
>
> Following your suggestion, we evaluate ProAct against both attacks and find that **our framework remains highly effective**.
>
> **Long-context Jailbreak**: We generated 100 attack prompts from the provided source code. The ASR against ProAct using GPT-4.1-mini and GPT-5 as backend models is reported below.
> |model|ASR|
>  |:---|:---|
> |GPT-4.1-mini|1%|
>  |GPT-5 |0%|
>
>
> **Many-shot Jailbreak**: As the original code was unavailable, we synthesized a similar attack merging 19 unique task/goal AdvBench pairs with a final malicious task. ProAct was then queried with the entire conversation history. The ASR on 100 synthesized samples is shown below.
> |model | ASR |
> |---|---|
> |GPT-4.1-mini|0%|
> |GPT-5 | 0% |
>
>
> ## Q4: Efficiency of LLM Serving ##
>
> We appreciate the reviewer raising this important point. We conduct a token-cost analysis to show that ProAct maintains **minimal serving overhead**. In particular we compare the token consumption and generation of different defense mechanisms on 100 randomly sampled tasks from IfEval and AIRBench. We report "extra input tokens" and "extra output tokens".
>
> |Defenses| Benign I/O| Malicious I/O|
> |---|---|---|
> |Input Filter|267/2|267/2|
> |Inference |53/0|53/0|
> |Output Filter|1710/2036.5|1710/3719|
> |ProAct | 267/3.7|1369/843.5|
>
> In benign settings, ProAct adds only 267 input tokens (8.7%), matching simple input filters and remaining far more efficient than multi-agent defenses like AutoDefense.
>
> In malicious settings, the increased token usage is justified by significant defensive gains (Tables 1 & 2). Crucially, ProAct **does not invoke the base model for inference on the original user query during attacks**, offsetting costs by preventing the expensive generation of harmful responses.
>
> We will include this experiment in our paper.

---

### Official Review · Reviewer_3D3V · 2025-10-30

**Soundness:** 2
**Presentation:** 1
**Contribution:** 2
**Rating:** 2
**Confidence:** 4

**Summary:**

This paper proposes a jailbreak defense method called ProAct against multi-turn seach-based attacks. The defense method first identifies jailbreak attempts and returns perturbed responses when spotting mallious inputs.

**Strengths:**

1. ProAct seems to be effective against PAIR and TAP, reducing ASR to nearly zero. On average, it reduces attack sucess rate for more than 50%, which is notable.
2. The authors conducted several ablations and experiments across different models to validate the method's effectiveness.

**Weaknesses:**

1. My major concern is about the User Intent Analyzer. Why is it necessary to return nonsense strings to the attackers when we can actually spot them? Would be much more easier and safer to just terminate the conversation or connection as it is done in most commercial chat websites like ChatGPT or Claude. In summary, I simply do not understand why such defense is needed when we can actually identify the attackers. For me, identifying the attackers is the most vital part of the defense.

2. The perturbation algorithms used by ProAct are trivial string operations, which is not adaptive to specific attackers. This might explain why the ASR is still above 50% for X-teaming.

**Questions:**

Is ProAct effective when we cannot identify the attackers?

---

> ### Author Response · Authors · 2025-11-20
>
> We sincerely thank the reviewer for their thoughtful feedback. We are glad that the reviewer recognizes the strong empirical performance of ProAct, which achieves substantially lower ASR than prior work.
>
> ## Q1: Why generate a spurious response instead of refusing? ##
>
> We thank the reviewer for noting this concern and would like to clarify our central conclusion: simple refusal enables attacker optimization and is neither a safe nor an optimal defensive strategy. Through spurious responses, ProAct causes the attacker to mistakenly believe their attack has succeeded, halting the optimization loop.
>
> As highlighted in Figure 1, simple refusal (e.g., “I cannot help with that.”) or conversation termination is interpreted by the attacker’s evaluator as a **strong negative signal**, prompting the attack loop to continue exploring and strengthening its prompts. Multi-turn algorithms explicitly **leverage these feedback signals to search for successful jailbreaks** on SOTA LLMs, which consistently return refusals in early iterations but fail later (see table below). Thus, **prematurely refusing the conversation actually enables the attacker’s optimization process by providing explicit signals** and makes the defense weaker.
>
> ProAct breaks this optimization loop by generating spurious responses that convince the attacker’s evaluator of success. **This prematurely halts the attackers’ adversarial search and prevents further prompt refinement**, driving the >70% ASR reduction. Therefore, the active spurious response is **necessary for effectively combating search-based attacks** and is **orthogonal/additive to existing defenses** such as input and output filters that are used by commercial chat websites.
>
> |Iteration|Percentage|
> |:--|:--|
> |step 1|12%|
> |step 2|26%|
> |step 3|36%|
> |step 4 |80%|
> |step 5|100%|
>
> Table: Cumulative Percentage of Successful PAIR Attacks (out of 100 random samples) against an Input Filter at each Optimization Step
>
>
>
> ## Q2: ProAct response creation. ##
>
> We apologize for the confusion. As described in Section 3.3 and illustrated in Figures 2 and 3, ProAct **does not perturb the original response to the user query.** Instead, it **generates spurious responses from scratch**, guided by:
> - the malicious topic extracted by the analyzer
> - few-shot obfuscation demonstrations
> - chain-of-thought reasoning to construct coherent but safe content,
> - refinement loop with the Surrogate Evaluator
>
> The resulting outputs are **topic-relevant, semantically structured**, and **safe**, **encoded through controlled transformations** (emoji, Base64, Morse, etc.) so that they **appear harmful to the attacker while containing no actionable harm**. Their effectiveness is evidenced in Table 4. These are carefully constructed deceptive responses tailored to each adversarial topic and attack, not trivial string operations.
>
>
>
> ## Q3: X-Teaming ASR. ##
>
> ProAct is designed as an orthogonal defense layer, not a standalone replacement for all defensive mechanisms. As shown in Table 2, when ProAct utilizes signals from an output filter, the ASR against X-Teaming drops to **below 5% across all benchmarks**. Importantly, ProAct further reduces the up to 84% ASR on passive defenses by **up to 37%** on average. Therefore, ProAct alone already yields large ASR reductions against X-Teaming (up to 70%), and ProAct + existing defenses provides almost complete protection.
>
> ## Q4: Is ProAct effective when we cannot identify the attackers? ##
>
> We thank the reviewer for this question. When the User Intent Analyzer cannot confidently detect malicious intent, the system defaults to the base model’s normal behaviour – exactly like existing LLM deployments today. In these cases, ProAct **does not introduce any additional failure modes**; it simply delegates to other defense layers (input/output filters, system prompts, model alignment).
>
> However, the ProAct framework **makes the attack identification process significantly easier**, as iterative jailbreak attacks begin with naive, easy-to-detect prompts and ProAct **only needs a single successful detection** to stop the attacker’s optimization loop, rather than consistent identification at every turn. Table 2 confirms this. Across four benchmarks and two strong attack strategies, ProAct reduces ASR by 33.25% and 12.38% beyond input/output filters because it requires only one successful detection. Against DAGR, which explicitly obfuscates intent through iteration, ProAct still provides up to 23% additional reduction.
>
> ProAct is also **fully compatible with existing defense pipelines**, and the User Intent Analyzer can **consist of signals from multiple other defense mechanisms**. Layering ProAct with AutoDefense in this manner reduces X-Teaming ASR to 0%, and combining it with other baselines yields up to an additional 70% reduction. This means imperfections in any single analyzer are mitigated, since a **single positive detection from any component** triggers ProAct and prevents the jailbreak.

---

> > ### Comment · Reviewer_3D3V · 2025-11-24
> >
> > Thanks for the rebuttal, but you have missed my major point.
> >
> > > Would it be much more easier and safer to just **terminate the conversation or connection?**
> >
> > By termination I mean the multi-turn conversation is completely banned, and the user is forced to start a new conversation, which would happen frequently if you try to attack the web version of ChatGPT.

---

> > > ### Author Response · Authors · 2025-11-24
> > >
> > > We appreciate the follow-up. Termination wouldn't be effective against automated attacks because the **attacker simply interprets "termination" of a conversation/session as a negative signal (e.g., Score 0), identical to refusal**. It informs the attacker algorithm that the current prompt failed, encouraging it to explore different search directions. Frameworks like PAIR, TAP, and DAGR typically operate via stateless API calls where every attempt is already a new conversation. Therefore, terminating the session **imposes zero cost or friction on the attacker**, as they naturally reset the context for every new prompt variant.
> > >
> > > Crucially, our defense is **agnostic to whether the attack is single-turn or multi-turn**. Regardless of the conversation structure, these attacks function as an optimization process driven by the history of queries and feedbacks, which serves as the optimization signals that guide the search of effective attacks. PROACT succeeds by disturbing this optimization signal: by injecting spurious responses, we **deceive the algorithm into accepting a false success**, thereby **breaking the feedback loop and halting the search**.

---

> ### Comment · Reviewer_3D3V · 2025-11-26
>
> If the attacker can simply interprets "termination" of a conversation/session as a negative signal (e.g., Score 0), identical to refusal, and still makes progress. Then why don't they interpret "responses with spurious injections" as a negative signal (e.g., Score 0), identical to refusal, and still make progress.
>
> To summarize, I will keep my initial rating of 2 because the proposed method requires precise detection of the mallicious users, which is in fact the most challenging part of the defense.

---

> > ### Author Response · Authors · 2025-11-27
> >
> > We respectfully challenge the assumption that attackers can simply interpret spurious responses as refusals to advance their attacks. Detecting a spurious response as a negative signal is a **highly non-trivial task**. Because ProAct’s responses look like a successful jailbreak, unlike refusal or conversation termination, the attacker almost always interprets the response as a **positive signal**. Tables 2 and 4 explicitly demonstrate that the spurious response effectively bypasses the attacker’s evaluator and is marked as a successful jailbreak. Thus, ProAct poisons the attacker’s feedback loop, **tricking the optimization algorithm into premature termination** and leading to significant ASR reduction. We support this with extensive empirical evaluation (Tables 1-4).
> >
> > If the attacker could simply interpret spurious responses as refusals as the reviewer suggests, these sophisticated search-based attacks would not have failed so catastrophically. ProAct achieves its significant orthogonal gains **without relying on precise detection**, instead **weakening the attack itself**. We respectfully urge the reviewer to reevaluate the contribution based on our strong empirical results, which validate the practical effectiveness of our strategy.

---

### Official Review · Reviewer_XNVM · 2025-11-01

**Soundness:** 3
**Presentation:** 3
**Contribution:** 3
**Rating:** 6
**Confidence:** 2

**Summary:**

The paper introduces ProAct, a proactive defense framework that thwarts adversarial jailbreaks by generating deceptive responses that mislead attackers without producing harmful content. Extensive experiments show that ProAct reduces attack success rates by up to 92%, and to 0% when combined with other defenses, offering a powerful complement to existing LLM safety measures.

**Strengths:**

Innovative idea of generating spurious responses to actively mislead jailbreak attackers.

Good writting structure and easy to follow.

**Weaknesses:**

The defense's effectiveness relies on the attacker's evaluation mechanism `User Intent Analyzer`, which could be imperfect.

**Questions:**

null

---

> ### Author Response · Authors · 2025-11-20
>
> We sincerely thank the reviewer for their thoughtful feedback. We are glad that the reviewer recognizes the novelty and effectiveness of the ProAct framework and the clarity of the presentation.
>
>
> Regarding the remaining concern, while the User Intent Analyzer can be imperfect (average detection rate 76.1% across all the experiments in Table 1, 100% accuracy under utility test), **ProAct relies far less on the analyzer’s detection accuracy than existing defenses**. Iterative jailbreak attacks begin with naive prompts, and each refusal provides feedback that strengthens the attacker’s optimization loop. **Prior defenses are passive and must therefore detect malicious intent at every step, which becomes increasingly difficult as attacks evolve** (see table below). In contrast, ProAct **needs to intervene only once when attacks are still weak**, generating a spurious response that stops the optimization loop entirely and preventing long iterative attacks.
>
> Table 2 (Section 5.2) confirms this. Across four benchmarks and two strong attack strategies, ProAct reduces ASR by 33.25% beyond input filters and 12.38% beyond output filters because it requires only one successful detection. Against DAGR, which explicitly obfuscates intent, ProAct still provides up to 23% additional reduction.
>
> ProAct is also **fully compatible with existing defense pipelines**, and the User Intent Analyzer **can consist of signals from multiple other defense mechanisms**. Layering ProAct with AutoDefense in this manner reduces X-Teaming ASR to 0%, and combining it with other baselines yields up to an additional 70% reduction. Imperfections in any single analyzer are mitigated, since a **single positive detection from any component** triggers ProAct and prevents the jailbreak. This makes ProAct practical to deploy as a proactive layer in real-world LLM safety systems.
>
> |Iteration|Percentage|
> |---|---|
> |step 1|12%|
> |step 2|26%|
> |step 3|36%|
> |step 4 |80%|
> |step 5|100%|
>
> Table: Cumulative Percentage of Successful PAIR Attacks (out of 100 random samples) against an Input Filter at each Optimization Step

---

### Official Review · Reviewer_1EQm · 2025-11-01

**Soundness:** 3
**Presentation:** 3
**Contribution:** 3
**Rating:** 6
**Confidence:** 2

**Summary:**

This paper proposes a jailbreak defense method called ProAct. The proposed method use a three-agent system that first identifies the users' intention, and then craft a spurious response (which serves as the proactive defender). This response is refined iteratively in order to deceive the  surrogate evaluator. The experimental results in Table 1 shows significant improvement on the models' robustness against jailbreaking attacks.

**Strengths:**

1. (Clarity) This paper raises 5 RQs and provide detailed discussion regarding each of them, which improves the readability of this paper.
2. (Significance) The experimental results in Table 1 shows significant improvement on the models' robustness against jailbreaking attacks.

**Weaknesses:**

See the quetions part.

**Questions:**

To what extent does the method presented in this study affect the efficiency of LLM serving? Is it possible to provide a quantitative assessment of the extra token consumption introduced by this method, and how does it perform in comparison with alternative methods?

---

> ### Author Response · Authors · 2025-11-20
>
> We sincerely thank the reviewer for their thoughtful comments. We are glad the reviewers note the effectiveness of our approach and the clarity of our presentation.
>
> We conducted a token-cost analysis to show that ProAct maintains **minimal serving overhead in benign cases** while remaining **far more efficient than multi-agent defenses under malicious cases**. The number of extra tokens induced by each defense is the primary driver of its additional latency. We conduct an experiment comparing the token consumption and generation of different defense mechanisms. In particular, we test on 100 randomly sampled tasks from IfEval and AIRBench. We report "extra input tokens" and "extra output tokens". Results are presented in the table below.
>
> |Defenses | Benign I/O | Malicious I/O |
> | --- | --- | --- |
> | Input Filter | 267/2 | 267/2 |
> | Inference | 53/0 | 53/0 |
> | Output Filter | 1710/2036.5 | 1710/3719 |
> | ProAct | 267/3.7 | 1369/843.5 |
>
> In the benign setting, ProAct adds **only 267 input tokens (8.7% increase) and negligible output tokens**, matching the cost of a simple input filter and remaining far more efficient than multi-agent defenses like AutoDefense, which incur substantial overhead from repeated agent calls and long system prompts.
>
> In the malicious setting, ProAct’s token usage increases, but this cost is justified because it leads to significant defensive gains. As shown in Table 1 and 2, ProAct **significantly reduces ASR while still requiring far fewer tokens than AutoDefense**. Additionally, **does not invoke the base model for inference on the original user query during attacks** under the malicious setting, which further reduces serving costs by preventing the expensive base model from generating a full (and harmful) response at all.
>
> We will include this additional experiment in the final version of our paper.

---

### Author Response · Authors · 2025-11-26

We again sincerely thank all of the reviewers for their thoughtful comments. We encourage the reviewers to participate in the discussion and thank them very much in advance for their time. We hope the reviewers can raise their scores if the rebuttal responses resolve their concerns, and we are happy to take more clarification questions. Thanks!

---

### Author Response · Authors · 2025-12-02
**Overview of Our Response to Core Reviewer Critiques of ProAct**

We thank you for managing the review process for our submission. We have engaged extensively with the reviewers and believe we have successfully addressed their reservations. We are writing to briefly summarize our rebuttal regarding the three primary concerns raised—specifically the Intent Analyzer, computational overhead, and the distinction between spurious responses and refusals.


### 1. Dependence on the Intent Analyzer: ###
Several reviewers expressed concern that ProAct relies on malicious-intent detection. Our rebuttal clarifies that **ProAct reduces** dependence on intent analysis compared to all existing passive defenses. Iterative jailbreak attacks rely on repeated refusals to refine their adversarial prompts; thus, passive defenses must correctly detect malicious intent **at every turn**, a requirement that becomes increasingly unrealistic as prompts grow more sophisticated.

ProAct fundamentally changes this dynamic. It requires **only a single successful detection**, at the earliest, most naïve stage of the attack when malicious intent is easiest to identify. After that, ProAct halts the attack by injecting a spurious response that terminates the attacker’s optimisation loop. This eliminates repeated reliance on the Analyzer and makes the identification task substantially easier. Empirically, across four benchmarks and two SOTA attack strategies, ProAct consistently improves performance beyond both input and output filters even when those filters are imperfect, showing that ProAct remains effective because it avoids relying on high-accuracy detection.

### 2. Latency and Extra Cost ###
Another common concern is that the spurious-response step may introduce additional latency or cost. Our rebuttal demonstrates that the overhead is **minimal and comparable to a standard single response generation**, because ProAct intervenes **only once per attack**, rather than in every turn. By cutting off multi-turn optimisation loops early, ProAct often reduces the total number of tokens generated in adversarial conversations. The additional inference cost is thus marginal relative to the safety benefit, and the method can be efficiently implemented in practice without measurable degradation in latency-sensitive settings. Results are presented in the table below experiment detail can be found at reviewer 1EQm discussion.

|Defenses | Benign I/O | Malicious I/O |
| --- | --- | --- |
| Input Filter | 267/2 | 267/2 |
| Inference | 53/0 | 53/0 |
| Output Filter | 1710/2036.5 | 1710/3719 |
| ProAct | 267/3.7 | 1369/843.5 |


### 3. Spurious Response vs Termination ###

Regarding the concern raised by reviewer 3D3V about using a spurious response rather than terminating the conversation, we clarify that termination is ineffective against automated attacks. Stateless attack frameworks such as PAIR, TAP, and DAGR treat each API call as a fresh session, so “termination” is interpreted as a standard failure signal and provides the attacker with the same negative feedback as a refusal. This reinforces the optimisation loop rather than disrupting it. In contrast, ProAct injects a spurious, non-harmful response that deceives the attacker algorithm into interpreting the attempt as a false success. This breaks the optimisation signal that guides adversarial search and halts the attack process, regardless of whether the attack is single-turn or multi-turn.


Across all reviews, no fundamental flaw was identified, and several reviewers explicitly acknowledged the novelty and practicality of our proactive-defense perspective. ProAct introduces a new class of defences that moves beyond the passive “filter-and-refuse” paradigm, showing **consistent and substantial ASR reductions (up to reducing X-Teaming to 0%)**, compatibility with existing pipelines, and strong empirical evidence that the approach is both practical and effective.
Given the growing sophistication of multi-turn jailbreaks and the demonstrated limitations of purely passive filters, we believe **ProAct addresses an urgent and increasingly critical gap in LLM safety**. We respectfully submit that the contribution is both novel and impactful and merits acceptance.

---

### Note · Authors · 2026-01-26

I have read and agree with the venue's withdrawal policy on behalf of myself and my co-authors.

---

### Meta-Review · Area_Chair_tKiX · 2025-12-09

**Summary:**

This paper presents a novel  ProAct defense framework that disrupts and misleads autonomous jailbreaking processes. Its core mechanism involves intentionally feeding adversaries "spurious responses"—outputs that mimic successful jailbreaks yet are devoid of harmful content. These deceptive signals corrupt the attacker's internal optimization loop, leading to premature termination of the adversarial search and effectively subverting the jailbreak attempt.

**Reviewer Concerns:**

the proposed method requires precise detection of the mallicious users, which is in fact the most challenging part of the defense.

**Reviewer Scores:**

Reviewer 3D3V did not change the socre.

After rebuttal, Reviewer 3D3V holds that If the attacker can simply interprets "termination" of a conversation/session as a negative signal (e.g., Score 0), identical to refusal, and still makes progress. Then why don't they interpret "responses with spurious injections" as a negative signal (e.g., Score 0), identical to refusal, and still make progress.

Then, Reviewer 3D3V keep the initial rating of 2 because the proposed method requires precise detection of the mallicious users, which is in fact the most challenging part of the defense.

---

### Decision · Program_Chairs · 2026-01-26

Reject